# A Stimulus-Responsive Polymer Composite Surface with Magnetic Field-Governed Wetting and Photocatalytic Properties

**DOI:** 10.3390/polym12091890

**Published:** 2020-08-21

**Authors:** László Mérai, Ágota Deák, Dániel Sebők, Ákos Kukovecz, Imre Dékány, László Janovák

**Affiliations:** 1Interdisciplinary Excellence Centre, Department of Physical Chemistry and Materials Science, University of Szeged, Rerrich Béla tér 1, H-6720 Szeged, Hungary; merail@chem.u-szeged.hu (L.M.); dagota13@yahoo.com (Á.D.); i.dekany@chem.u-szeged.hu (I.D.); 2Interdisciplinary Excellence Centre, Department of Applied and Environmental Chemistry, University of Szeged, Rerrich Béla tér 1, H-6720 Szeged, Hungary; daniel_sebok@yahoo.com (D.S.); kakos@chem.u-szeged.hu (Á.K.)

**Keywords:** stimulus-responsive, composite surface, magnetoresponsive, switchable wetting, photoreactive

## Abstract

With the increasing demand for liquid manipulation and microfluidic techniques, surfaces with real-time tunable wetting properties are becoming the focus of materials science researches. In this study, we present a simple preparation method for a 0.5–4 µm carbonyl iron (carbonyl Fe) loaded polydimethylsiloxane (PDMS)-based magnetic composite coating with magnetic field-tailored wetting properties. Moreover, the embedded 6.3–16.7 wt.% Ag-TiO_2_ plasmonic photocatalyst (d~50 nm) content provides additional visible light photoreactivity to the external stimuli-responsive composite grass surfaces, while the efficiency of this photocatalytic behavior also turned out to be dependent on the external magnetic field. The inclusion of the photocatalyst introduced hierarchical surface roughness to the micro-grass, resulting in the broadening of the achievable contact and sliding angle ranges. The photocatalyst-infused coatings are also capable of catching and releasing water droplets, which alongside their multifunctional (photocatalytic activity and tunable wetting characteristics) nature makes surfaces of this kind the novel sophisticated tools of liquid manipulation.

## 1. Introduction

Stimuli-responsive polymers and polymer composites are state-of-the-art products of modern day materials science. They have several fields of application, ranging from pH-responsive intelligent drug delivery and release systems [1] to photo- or electroresponsive actuators [2,3]. While in the case of 3D materials, there are approaches to influence properties such as swelling degree, bulk density, shape, and size [4], in the case of 2D materials, surface properties could be the subject of stimuli-responsiveness. These properties include surface functionality, charge excess, and roughness [5,6]: manipulating these by external stimuli could result in the reversible change of wetting properties, which is beneficial in microfluidics and other liquid manipulation applications.

Although, many magnetic surfaces with superhydrophobic properties (Θ ≥ 150°) have been developed so far, including magnetic sponges for water-oil separation [7] and hybrid coatings for electromagnetic shielding applications [8], magnetic field-induced wettability transitions are still less studied than systems with electric field [3,9], light [10], temperature [5,6,11], and pH-responsivity [12].

While liquid manipulation by magnetic field is possible through dispersing particulate materials in the liquid itself [13], the enhancement of solid surfaces with magnetic fillers offers a better alternative because in that case the physical and chemical properties of the wetting liquid phase can remain unchanged. If the elasticity of the base material is high enough to be capable of reversible deformation or spatial organization while keeping the magnetic particles inside as they align in the direction of the external magnetic field, considerable changes in surface roughness and wettability can occur [14]. In most of the few existing approaches, the applied 2D magnetorheological elastomers are polydimethylsiloxane (PDMS) based due to its chemical resistance, remarkable but easily adjustable (e.g., with crosslinking agents or plasticizers such as silicone oil) elasticity, and initially hydrophobic nature, and the magnetic filler material is hydrophobized particulate ferromagnetic carbonyl Fe in many cases [14,15]. As these surfaces mostly consist of uncured PDMS or contain high amounts of plasticizer to achieve broad wettability ranges, their applications are limited due to their stickiness and plasticity.

More promising types of tunable coatings are the anisometric 3D magnetic grasses or pillars [16,17]: these also contain the same materials (PDMS and fine Fe particles), but during their preparation the crosslinking of the PDMS takes places in an external magnetic field which directs the magnetic particle-content (and therefore the ambient PDMS matrix) along the force field lines. The resulting crosslinked products are elastic grass hairs with magnetic field-directed orientation and field-strength dependent stiffness. Yang and their coworkers prepared such coatings by applying a magnetic-field directed spray-coating technique [17]: as a permanent magnet was placed under the substrate, the solids-content of the spraying dispersion gathered along the force field lines and the crosslinking of PDMS resulted in the formation of the anisometric surface structures with two reported wetting states. Without external magnetic field, the elastic strands can be aligned randomly, allowing waterdrops to penetrate the grass while providing a contact area high enough to let the adhesive forces keep the coating wetted (line contact), but in the magnetic field the strands stiffen: the contact between the grass and the droplets becomes pinning (pinning contact), resulting in decreased adhesion, and moreover, superhydrophobic behavior is also achievable. Thanks to these reversible transitions and the achievable broad contact and sliding angle ranges, these coatings are excellent tools for liquid manipulation scenarios, as they can pick up and release water droplets without nearly any loss of liquid [17].

To broaden the range of surface functionalities, hydrophobic polymer-based surfaces can also be enhanced with photocatalyst particles, resulting in photocatalytic and even antimicrobial properties; besides, they can also be utilized to induce changes in wetting properties through surface roughness modification [18]. In our previous works, spray-coated hydrophobic fluoropolymer- and PDMS-based layers were enhanced with visible light-active plasmonic Ag-TiO_2_ photocatalyst nanoparticles to obtain bifunctional (superhydrophobic and visible light-active) [19,20] and trifunctional (with extra self-healing ability) coatings [21]. Despite there being no examples of combining the magnetic field dependent wetting and photocatalytic properties in the literature so far, these multifunctional surfaces may open new perspectives for sophisticated liquid manipulation techniques.

In this study we succeeded in infusing the beneficial photocatalytic and external stimuli responsive wetting properties into one coating: we present the preparation and characterization of visible light-active Ag–TiO_2_-enhanced magnetic grass surfaces with composition, magnetic field strength and direction-dependent wetting and photocatalytic properties. We also show that the photocatalytic efficiency of this novel coating material was adjusted by an external magnetic field through the wetting properties of the layer.

## 2. Materials and Methods

### 2.1. Preparation of Magnetic Grass Coatings

As an elastic and hydrophobic polymer matrix material, two-component Elastosil C1200 PDMS elastomer (Wacker, Munich, Germany) was used: component A contained the silicone-hydride prepolymer and the vinyl-terminated crosslinker (component A will be referred as crosslinker), amd component B (later referred as catalyst) contained the prepolymer alongside the Pt-complex catalyst for the curing reaction. The synthesis of the used visible light-active Ag-TiO_2_ photocatalyst (d_mean_~50 nm) nanoparticles is described in our previous publications [22,23]. During the composite film preparation process, 1.5–1.5 g portions of the two PDMS components were dissolved in 9 mL of toluene (ar.; Molar Chemicals, Halasztelek, Hungary); then 4.5 g of carbonyl Fe particles (d = 0.5–4 µm; Sigma-Aldrich) were dispersed in the solution alongside 0.5, 1.0 or 1.5 g Ag-TiO_2_ (6.3, 11.8 and 16.7 wt.% nominal photocatalyst content, respectively) followed by a 1 min ultrasonication. The resulting dispersions were sprayed on clean 7.6 × 2.6 cm^2^ glass microscope slides, while a permanent magnet with a surface magnetic flux density of 0.30 T or 0.35 T was placed right behind the substrate. The components were then deposited on the slides. For contact angle measurements, the grass heights were uniformly set to 3 ± 0.1 mm, but in the case of photocatalytic tests, grass specific masses were uniformized and set to 63.3 ± 1.6 mg/cm^2^. The samples were cured at room temperature for 3 h without the magnet being removed. As blank samples for the photocatalytic tests, 63.3 mg/cm^2^, 7.6 × 2.6 cm^2^ plain spray-coated composite layers were also prepared without magnetic particles (16.7 wt.% Ag-TiO_2_ and 83.3 wt.% PDMS).

### 2.2. Methods of Characterization

The morphologies of the particles and the layers were examined by field emission scanning electronmicroscopy (SEM−Hitachi S-4700 microscope), while applying a secondary electron detector and 10 kV acceleration voltage.

To study the morphologies of the photocatalyst nanoparticles, transmission electron microscopy (TEM) measurements were performed using a FEI Tecnai G2 20 X-TWIN microscope with a tungsten cathode operated at 200 kV.

The X-ray computed tomography (CT) analysis of the magnetic grass was performed with a multiscale X-ray nanotomograph (Skyscan 2211, Bruker, Kontich, Belgium), equipped with a CCD camera. The applied acceleration potential was 110 kV, while the pixel resolution was 2 μm. The grass height was determined by the help of X-ray CT images and an internal scalebar of the used software.

The specific surface area sof the Ag-TiO_2_ photocatalyst and the magnetic grass were determined by nitrogen adsorption at 77 K by a Micromeritics gas adsorption analyser (Gemini Type 2375), applying the BET method.

The elastic properties of the PDMS-based polymer binders were studied during oscillatory viscometry measurements (Anton Paar Physica MCR 301) at 25 ± 0.1 °C, applying the PP20 probe with 1–1.2 mm gap width. The storage (G′) and loss (G″) moduli were measured while constantly increasing the load on the samples (in the 0.01–1000% deformation range), with the applied angular frequency of 10 s^−1^.

To measure the apparent static contact angles (Θ) of the layers, a drop shape analysis system was applied (EasyDrop, Krüss GmbH, Hamburg, Germany) at 25.0 ± 0.5 °C. A 8 ± 1 µl drop of water was made on the sample with the use of a syringe, equipped with a stainless steel needle. Using the CCD camera of the goniometer, the drop contour of the registered photo was mathematically described by the Young–Laplace equation using DSA100 software, and the Θ was determined as the slope of the contour line at the three-phase contact point. During contact angle measurements, the magnetic field dependence of Θ values were measured with the help of a 0.30 T bar magnet, placed under the samples to provide permanent external magnetic field and vertical stiffening of the grass. To align and stiffen the strands horizontally, the magnet was turned by 90°. To measure sliding angles, our drop shape analysis system was enhanced with a custom tilting cradle and a built-in goniometer.

The magnetic flux density values of the applied magnets were measured using an AlphaLab (Salt Lake, UT, USA) Model GM2 DC Magnetometer.

The photocatalytic efficiency of the magnetic grass coatings was studied at the S/L-interface. The layers were placed in plastic Petri dishes with the diameter of 9.0 cm; then they were carefully immersed in 50 mL of 2 mg/L (6.25 μM) aqueous solution of the Methylene Blue model organic pollutant, and were left in dark for 30 min to reach the adsorption equilibrium. The immersed plates were then illuminated for 300 min by a blue light LED-lamp (λ_max_ = 405 nm; General Electrics, Budapest, Hungary) [20], placed 5 cm above the gently agitated (50 rpm) samples. During the experiments the actual concentration of Methylene Blue was determined by regularly taking 3.2 mL samples of solution using an automated pipette and applying a Red Tide SHIMADZU UV-1800 spectrophotometer. The decreasing absorbance values at λ = 660 nm were followed according to a previously recorded calibration curve. After the 1 min spectra recordings, the sampled volumes were poured back into the experimental solution. The photodegradation tests were repeated in (0.30 T) external magnetic field, as well (with vertically-stiffened grass). All photocatalytic tests were repeated 3 times.

## 3. Results

### 3.1. Characterization of the PDMS Matrix Material

The wetting of a solid surface is highly influenced by its roughness [24,25]. In the case of magnetic grass composites, we manipulate the surface roughness through directing the strands in an external magnetic field: the grass stiffens when the field, perpendicular to the substrate is turned on, resulting in a rough surface, decreased contact area, and adhesion between the droplet and the grass, which means higher contact angle values [17]. To provide reversible shaping and wetting transitions, the elasticity is a key property of these composites [15,16,17].

Due to its easily tunable elasticity and hydrophobic nature, PDMS was used as a matrix material during the preparation of magnetoresponsive composite coatings with tunable wetting properties. Upon mixing the two components of the PDMS, the curing took place according to the scheme of Figure 1a [26] for 3 h at room temperature and in ambient air. The crosslinking reaction took place between the silicone hydride and the vinyl-terminated silicone oligomer components through coordinative-polymerization by a Pt-complex catalyst [27]. We examined the elastic (storage moduli; *G*′) and viscous (loss moduli; *G*″) properties of evolving PDMS samples with different amounts of crosslinker during oscillatory viscosity measurements: Figure 1b shows the characteristic modulus vs. deformation curves of PDMS-samples with different crosslinking densities. As Figure 1c shows, the elasticity maximum was reached at 1:1 (mass-to-mass) ratio, and by increasing either the crosslinker (component A) or the catalyst (component B) of the PDMS, the measured values drastically decreased; i.e., excess of the catalyst or the crosslinker resulted in more liquid-like characteristics.

To achieve stimuli-responsive surfaces with reversible shape and wetting transitions, high elasticity is essential: as the 1:1 composition possessed the highest *G*′ and *G*″ values and hence the most elastic behavior among all samples, this polymer mixture was applied in our further experiments.

### 3.2. Preparation and Surface Morphologies of Magnetoresponsive Grass Coatings

To form elastic coatings with dual magnetoresponsivity and photoreactivity, the elastic PDMS matrix was loaded with 59.2–50.0% carbonyl Fe microparticles and 6.3–16.7% plasmonic Ag-TiO_2_ nanoparticles, respectively. As the TEM image of Figure 2 shows, the nearly spherical photocatalyst particles are nanosized (d~50 nm) and contain Ag nanodots on their surface (0.5 wt.% Ag content), which makes their excitation with visible light possible (λ_Ag;max_ ~ 450 nm) [28]. However, the spherical carbonyl Fe particles are microsized with relatively broad (*d* = 0.5–4 µm) size distribution.

The preparation process of the magnetic grass composites is a combination of spray-coating and magnetic field-directed self-assembly on glass substrates, which is based on the fact that magnetic particles also direct each other by their own magnetostatic forces in external magnetic field: such systems can reach their free-energy minimum upon the formation of ordered structures [17,29], which in this case are the magnetic Fe/PDMS pillars.

Spray-coating techniques in general have many parameters that influence product quality [30]. The size of the nebulized droplets is dependent on the viscosity and the surface tension of the spraying dispersion [31], but nozzle geometry, the applied gas pressure, and the sprayed amounts are also determining factors [31,32]. As in the case of magnetoresponsive grass coatings, the provision of adequate strand density and geometry (thickness and height) is vital to achieving reversible wetting transitions; in other words, the distances between the single strands have to be small enough to produce the desired non-wetted rough surface, which can result in pinning contact between the grass and the droplets. These vital parameters are mostly affected by the magnetic flux density, the distance between the substrate and the permanent magnet, and the droplet size. Higher flux density generally leads to denser grass [17], but as the self-assembly is also driven by the magnetostatic interactions between the single particles, the size of the nebulized droplets is more important as thinner strands attract less nebulized particles than the ones formed upon the drying of bigger droplets.

Yang et al. extensively studied the effects of these parameters on product quality and found their proper ranges [17]. According to their study, the suitable spraying composition to achieve reversible wetting transitions requires carbonyl Fe, PDMS and toluene in a (4.5:3:7.8) mass-to-mass ratio, respectively, and the appropriate amount of sprayed dispersion to reach optimal grass height (~2 mm) was 3–4 mL. Reference [17] specifies that the provision of similar heights is vital in order to achieve contact areas large enough to increase the adhesion of the randomly-oriented grass (0 T magnetic field), although—according to our observations—even higher grasses may suffer from the inability to reversible wetting transitions.

Before enhancing these grass coatings with photocatalytic properties, one must consider that as the features of these systems already depend on many parameters, the addition of another particulate (photocatalyst) component may imply further optimization on the preparation process.

However, according to our experiments the grass coatings can be infused with the Ag-TiO_2_ photocatalyst without losing their reversibly switchable wetting characteristics. During the preparation process, the proportions of Yang’s original composition formula were kept (4.5:3:7.8 = carbonyl Fe:PDMS:toluene; mass-to-mass ratio) [17]—besides the addition of increasing amount of photocatalyst nanoparticles into the spraying dispersion. As the SEM images of grasses prepared in 0.30 T magnetic field in Figure 3 show, the surface of the initially regular strands become rougher as the nanoparticle content is increased. As it was presented in our previous publications, the spherical-like surface structures that appear are characteristic of the spray-coated composite layers of the Ag-TiO_2_ photocatalyst [20,21]. The dispersibility of the hydrophilic photocatalyst particles into the organic spraying medium (toluene) is low; hence, the limit of the nominal photocatalyst content was set to ~16.7 wt.% to avoid extreme aggregation, which would cause the instability and thus the undesired vulnerability of the composites. Above this photocatalyst content, the adhesion of the grass to the glass substrates was dissatisfactory and the switching between the wetting states could not be repeated as the grass was pulled off in the proximity of magnetic field. However, the SEM images in Figure 3 show moderate aggregation, resulting in the formation of thicker strands and less dense grass alongside the increase in surface roughness. As the dispersibility of carbonyl Fe in toluene and PDMS is higher, the aggregation and the formation of surface irregularities are less considerable.

### 3.3. Wetting Properties of Magnetic Grass Coatings

To quantitatively measure the magnetic field-dependent wetting characteristics of the magnetic grass coatings, contact angle (sessile drop technique) and sliding angle measurements were applied in the presence or without a 0.30 T magnetic field, directed vertically or horizontally to the substrate. Both series of the prepared samples (sprayed in 0.30 T and 0.35 T magnetic fields) were studied: the detailed comparison can be seen in Figure 4.

As we have already shown in the previous chapter, the addition of Ag-TiO_2_ photocatalyst nanoparticles into the magnetic grass coatings can preferably increase the overall surface roughness of these systems. It is generally known, that lotus leaf-like hierarchical (micro- and nanoscale) surface roughness can amplify the initially hydrophobic or hydrophilic wetting character of a flat surface [25]. As Figure 4 shows, due to this increase in roughness, the vertically-stiffened pinning grasses (in the middle) show increasingly hydrophobic character, and even superhydrophobicity (150.9° < θ < 163.6°) can be achieved with higher amounts (>6.3 wt.%) of Ag-TiO_2_. Without an external magnetic field, however, the strands are randomly oriented (on the left; see schematic representation and the corresponding droplet images of Figure 4), and thus can be compressed by the sessile droplets. In this case the opposite trends were observed: the contact angles showed a little decrease as the photocatalyst content increased, which can be attributed to the roughness-induced increase of surface area in line contact mode and the decrease of grass density due to aggregation. However, during the evaluation of these data, it is important to note that the presented contact angle values are apparent as the droplets did not actually spread on the surface but penetrated the grass, which practically means that less dense and taller grasses allow deeper penetration for the droplets; therefore, they provide increased contact area and adhesion between the grass and the droplet [17]. This phenomenon can also be obviously observed the X-ray computer tomography images of Figure 5, in which the liquid drop shapes are visible thanks to the better contrast. The images clearly indicate that the randomly-oriented state of the surface allows larger wetting area than the vertically-stiffened state.

Grass density and thus indirectly the wetting characteristics can also be adjusted by changing magnetic flux density: Figure 4 also shows that the achievable contact and sliding angle ranges of grass surfaces prepared in 0.35 T magnetic field are broader that the ones sprayed in 0.30 T with the same composition. In the vertically-stiffened state (middle), the increases in photocatalyst content and roughness generally resulted in increasing contact angle values, while the line contact mode of random orientation (left) had smaller contact angles due to the increased contact surface area and adhesion [17].

Moreover, the randomly aligned grasses have good adhesion to water droplets with sliding angles of 180° (Figure 4, left), independent from the photocatalyst content (in the 0–16.7 wt.% range), but in vertically aligned state (middle) the sliding angle values gradually decrease with increasing photocatalyst content; this also proves the influence of surface roughness over hydrophobicity.

In addition to these, we studied the wetting of a third possible grass alignment: the contact angles were also measured on lying grass surfaces with horizontally stiffened strands. To reach this state, we applied 0.30 T magnetic field with force field lines parallel to the surface of the substrate. As our results show (Figure 4, on the right), this grass alignment is very similar to the one of the vertically-stiffened grass (middle) in terms of wetting properties: the photocatalyst-enhanced rough sides of the strands possessed contact angles resembling the values of the vertically-stiffened state (Figure 4, in the middle). This behavior indicates similarities in the spacings between the single vertically-stiffened hydrophobic strands and between their surface microstructures.

After all, it is important to note that the achievable contact and sliding angle ranges were barely dependent on the magnetic field (0.30–0.35 T range), applied during sample preparation (except in the case of randomly-oriented grasses); however, there are examples of broader sliding angle ranges (e.g., 8–180°) in the literature where the applied magnetic flux density was slightly higher (4.5 T) [17].

Thanks to the achievable broad sliding angle ranges and the fast-response reversible switching, the magnetic grass coatings with each composition can be used to pick up and release water droplets. As it can be seen in Figure 6 and in the Appendix A, when no external magnetic field is present, the randomly aligned strands with sliding angle of 180° (Figure 4) are able to catch and pick up a water droplet from a superhydrophobic surface, but when a 0.30 T permanent magnet is placed over the sample, the grass stiffens and the droplet falls off due to the suddenly increasing contact angles (163.9 ± 2.9°) and decreasing sliding angles (32.0 ± 7.4°) (Figure 4). The process can be repeated many times, and as this behavior is not sensitive to the composition (0–16.7 wt.% Ag-TiO_2_) and the self-assembly assisting magnetic field (0.30–0.35 T) in the studied ranges, these robust systems offer versatile liquid manipulation capabilities and the subsequently presented visible light photocatalytic activity, as well.

Besides the previously presented static contact angle measurements (Figure 4.), the magnetic field-adjustable wetting of the composite layer caused by the controllable surface orientation of the strands was also presented in dynamic mode. Images presented in Figure 7 captured from Appendix A show that the position of a non-wetting region of the composite surface can be arbitrarily changed by moving a 0.30 T magnet bar below the glass substrate.

### 3.4. Magnetic Field-Tailored Photocatalytic Properties

In this chapter we study the effects of magnetic field tailored wetting properties on the photocatalytic behaviors of the composites. The photocatalytic efficiencies of the applied plasmonic Ag-TiO_2_ semiconductor photocatalyst and its polymer based (PDMS, fluoropolyer, etc.) composites were proved in various scenarios, including the degradation of model organic pollutants at the S/G- and at the S/L-interfaces and antibacterial tests [20,21,22,23,28]. In contrast to the UV-active TiO_2_ (*E*_g_ = 3.2 eV), the nanosized particles with a mean diameter of ~50 nm (Figure 2) possess band gap energies of ~2.9 eV due to the presence of the *d* = ~5 nm plasmonic Ag nanodots on their surface (0.5 wt.% Ag content), which makes their excitation with visible light possible (λ_Ag;maxn_ ≈ 450 nm) [28]. At the semiconductor/liquid-interface, reactive oxygen species (e.g., hydroxyl radical) form during illumination [33], which can completely mineralize organic compounds due to their high oxidative potential. The overall photocatalytic activity of these semiconductor composite surfaces also depends on many other factors: their surface roughness, porosity, wettability, and adsorption affinity towards the oxidizable molecules are equally important [34,35]. Methylene-blue (MB) is a widely applied model pollutant in the evaluation of the photocatalytic activity of photoreactive coatings, as its concentration can easily be determined through recording the VIS absorption spectra of the colorful MB solutions [35]. If the pH is neutral and enough oxygen is present in the solution, only the mineralization of the MB molecules contributes to the decolorization, which allows a relatively exact determination of photocatalytic efficiency [35]. The emission intensity of the exciting light source should also be low in the λ = 350–480 nm range to avoid direct MB photolysis [35]: our LED-lamp fulfilled this requirement with an emission maximum of λ = 405 nm. As TiO_2_ has a PZC value of pH ≈6, the adsorption of the positively charged MB molecules has good affinity to the negatively charged catalyst surface at pH ≈7 [36].

However, the wetting properties of a photoreactive layer can also significantly affect photoactivity. We previously presented that a rough, non-wetting superhydrophobic coating, composed of 80 wt.% Ag-TiO_2_ and 20 wt.% fluoropolymer can degrade only ~20% of the Methylene Blue in aqueous solution after 90 min illumination with a blue-light LED lamp (λ_max_ = 405 nm) [20]. According to our previous EDX-studies, the similar spray-coated Ag-TiO_2_/polymer composite surfaces contain homogenously dispersed photocatalyst particles, which results in the reported considerable photocatalytic activity [21].

In the case of our photocatalyst-enhanced grass samples with 63.3 ± 1.6 mg/cm^2^ specific mass and 16.7 wt.% nominal photocatalyst content, the hydrophobic, i.e., non- wetting nature, resulted in similarly moderate decomposition efficiencies. Moreover, as Figure 8 shows, this photocatalytic efficiency was also influenced by the external magnetic field-assigned wetting properties: when no magnetic field is present, the randomly-oriented grass collapses under the liquid medium (Figure 7), resulting in higher contact area and thus higher photocatalytic efficiency (the initial MB concentration decreased by 47.6 ± 5.1% after 300 min irradiation), and when a 0.30 T magnet was placed under the sample during illumination, the contact area between the vertically-stiffened grass and the liquid reduced due to the resulting pinning contact. Figure 8a shows the absolute concentration changes with the corresponding adsorbed amount of MB, while Figure 8b shows the relative (c/c_0_) changes in MB solution concentration during LED light illumination.

The determined photocatalytic efficiencies of the grass coatings were 42.1 ± 3.5% (random orientation) and 22.4 ± 3.3% (vertically stiffened), respectively, after the 300 min illumination periods (Figure 8b). According to the synthesis conditions, the nominal Ag-TiO_2_ content of each sample was 0.208 g; thus, the photocatalytic measurement was repeated with this photocatalyst content but without PDMS for reference. It can be seen that the pure Ag-TiO_2_ photocatalyst nanopowder (without PDMS matrix) completely decomposed the MB-content of the solution under the same experimental conditions. This was due to the increased accessibility of the hydrophilic photocatalyst particles with BET surface area of 53.5 m^2^/g; however, the measured data clearly indicate that the PDMS-based composite with much lower BET surface area (0.15 m^2^/g) also showed obvious photoreactivity, which was also affected by the wetting properties, as was previously described. As it can be seen in Figure 8a, the pure photocatalyst particles with higher surface area and more hydrophilic nature adsorbed more MB during the dark period of the experiments. Figure 8 also shows that the photolysis of MB was insignificant, indicating that only the photocatalytic process on the grass surface caused the decolorization of the MB solutions.

As a comparison with the magnetic composites, photoreactive layers without magnetic particles were also prepared, consisting only 16.7 wt.% Ag-TiO_2_ and PDMS. The morphological, wetting, and photocatalytic behaviors of similar layers were discussed in our previous publications [20,21]. Unsurprisingly, the layers showed significant photoreactivity in this case as well, with 52.0 ± 1.8% MB decomposition-efficiency (Figure 8).

Thanks to the presented magnetic field-responsive wetting and photocatalytic characteristics, the prepared grass coatings may seek further applications, not only as sophisticated liquid manipulation tools but utilizable solutions when it comes to the mineralization of organic pollutants.

In our previous publications, we described other Ag-TiO_2_-containing composite surfaces with systematically changing, composition-dependent wettability and adsorption affinity towards model pollutants with different polarities [20,21]. The now-presented composites with real-time tunable wetting properties potentially offer the same behavior, and they are, for instance, good candidates for such water-treatment and cleaning applications, in which the polarities of the contaminating chemical species change on a wider scale. To prove this theory, further investigations on the photocatalytic behavior of these magnetic composites against different kinds of model pollutants are planned as well.

## 4. Conclusions

With the increasing popularity of microfluidic and liquid micro-manipulation techniques, there is an emerging necessity for developing new functional surfaces. To address this need, PDMS- and carbonyl Fe-based composite grass coatings with magnetic field-sensitive wetting properties and visible light-photoactivity were prepared, by applying a magnetic, field-assisted, spray-coating, self-assembly technique. The coatings possess variable contact angle and sliding angle ranges depending on the external magnetic field strength and the Ag-TiO_2_ photocatalyst content, which introduces surface nano-roughness and surface irregularities to the magnetic grass systems. The achievable wetting ranges of the prepared samples (with or without external magnetic field) were found to be expanding with the increasing photocatalyst content (0–16.7 wt.%) and during the spray-coating process.

The inclusion of photocatalyst nanoparticles also provided visible light photoactivity to the coatings: during blue LED illumination (λ_max_ = 405 nm) the layers could decrease the concentration of the Methylene Blue (MB) model organic pollutant (c_0_ = 2 mg/L) at the solid/water-interface. Moreover, the photocatalytic efficiencies turned out to be dependent on the external magnetic field as well: without an external magnetic field, the relatively well-wetted (Θ = 56.5 ± 0.5°), randomly-oriented grass decreased the initial MB concentration by 42.1 ± 3.5% in 300 min; however, the photocatalytic efficiency in the case of the non-wetted (Θ = 163.6 ± 2.9°), vertically-stiffened grass (in 0.30 T magnetic field) was only 22.4 ± 3.3%.

The external magnetic field-induced reversible wetting transitions of the presented coatings made droplet transportation with minimal loss possible: small water droplets could be picked up and released with all grass compositions by turning off and on the external magnetic field, respectively.

These intelligent bifunctional surfaces were proven to be versatile tools of liquid manipulation and may find potential applications in the near future.

## Figures and Tables

**Figure 1 polymers-12-01890-f001:**
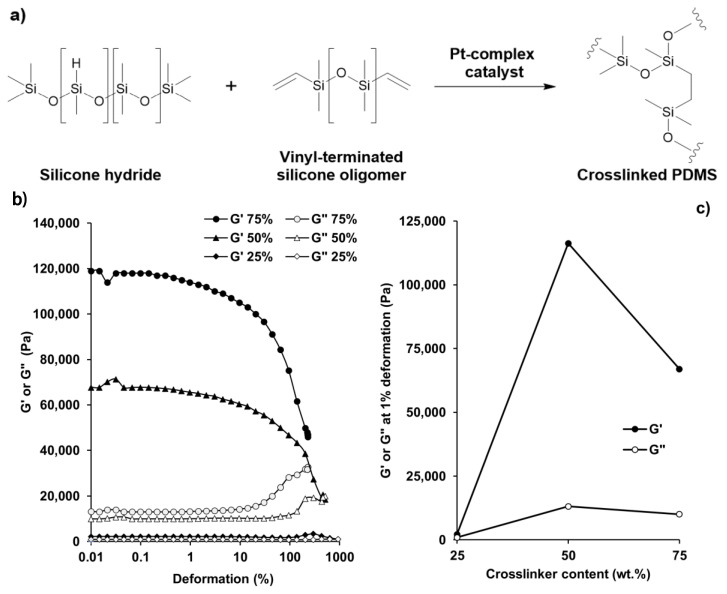
Schematic representation of PDMS crosslinking. (**a**) The measured storage (*G*′) and loss moduli (*G*″) of PDMS matrices with different wt.% crosslinker contents (%) as a function of applied deformation (**b**); and the effects of crosslinker content on the dynamic viscoelastic properties of the samples; *G*′ and *G*″ values at 1% deformation (**c**).

**Figure 2 polymers-12-01890-f002:**
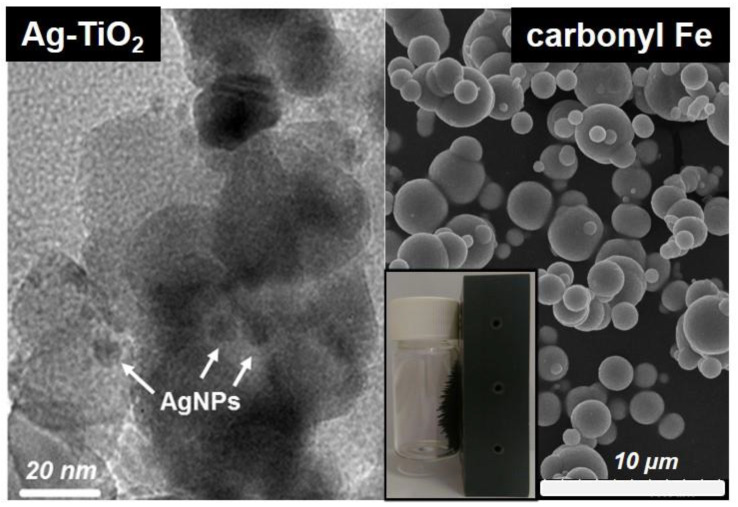
TEM image of the plasmonic Ag-TiO_2_ photocatalyst particles with surface AgNPs and a SEM image of Ag-TiO_2_ and carbonyl Fe particles with magnetic properties; the magnetic property was demonstrated by grabbing the Fe-containing glass vial with a permanent magnet (inset image).

**Figure 3 polymers-12-01890-f003:**
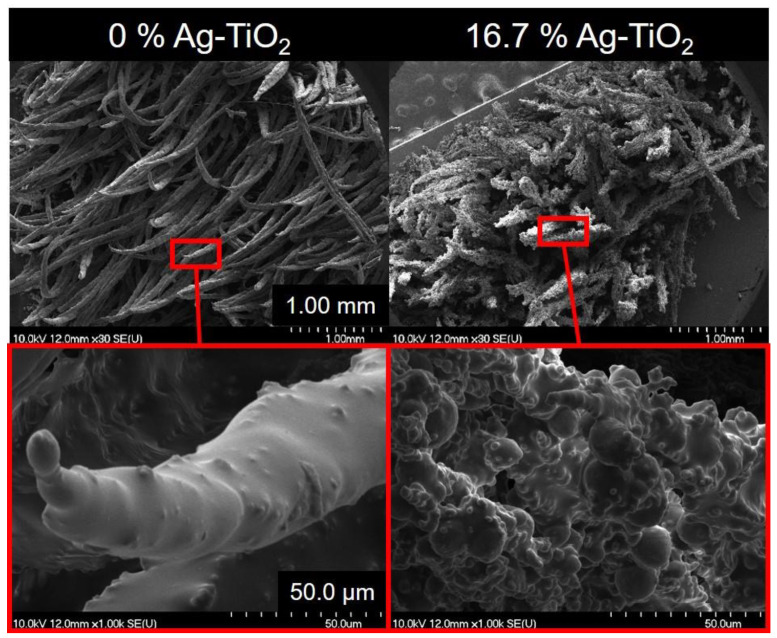
SEM images of magnetic grasses with 0 and 16.7 wt.% nominal Ag-TiO_2_ content cured under 0.35 T magnetic field.

**Figure 4 polymers-12-01890-f004:**
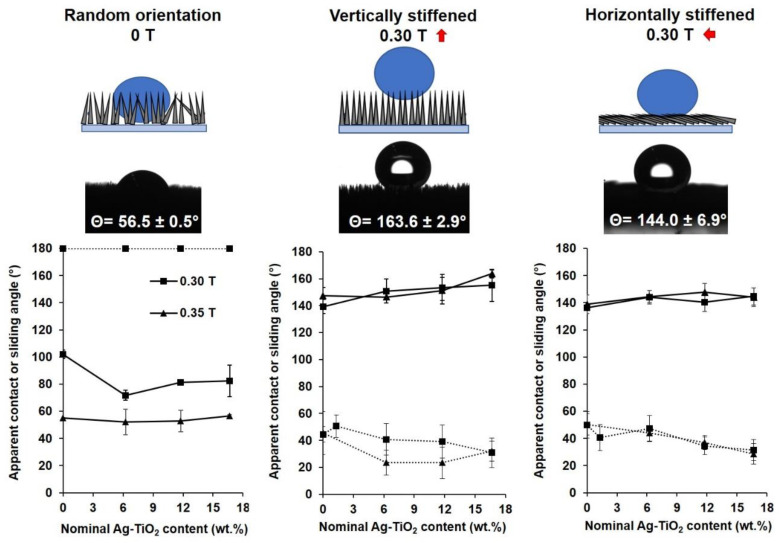
Schematic representations of the three wetting states of Ag-TiO_2_-enhanced magnetic grass coatings with the corresponding drop photographs on grass samples with 16.7 wt.% Ag-TiO_2_, prepared in 0.35 T magnetic field (**top**) and apparent water contact (**continuous lines**) and sliding angles (**dotted lines**) on magnetic grasses at the corresponding wetting states, cured under 0.30 T or 0.35 T magnetic fields as a function of the nominal Ag-TiO_2_ content.

**Figure 5 polymers-12-01890-f005:**
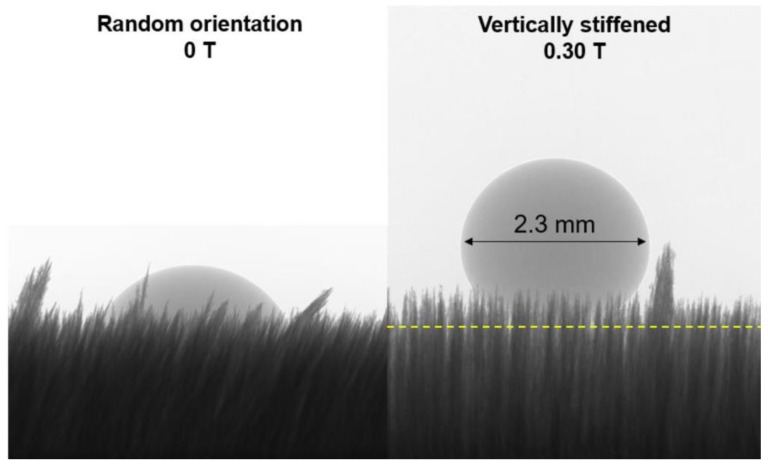
X-ray CT images of water drops on randomly-oriented (**left**) and vertically-stiffened (**right**) magnetic grasses with 16.7 wt.% nominal Ag-TiO_2_ content.

**Figure 6 polymers-12-01890-f006:**
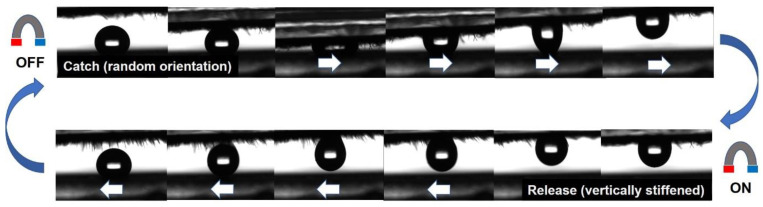
CCD images of water droplet catch and release by magnetic grass coatings with 16.7 wt.% Ag-TiO_2_ content.

**Figure 7 polymers-12-01890-f007:**
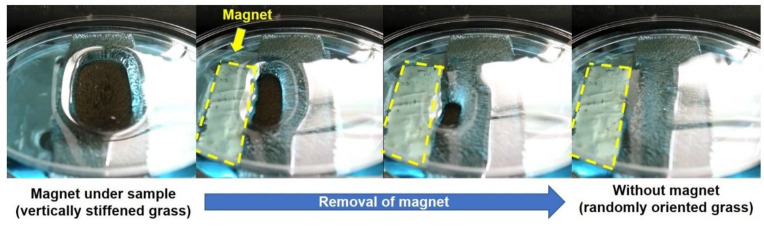
Switchable wetting characteristics of magnetic grass coatings in contact with bulk aqueous phase.

**Figure 8 polymers-12-01890-f008:**
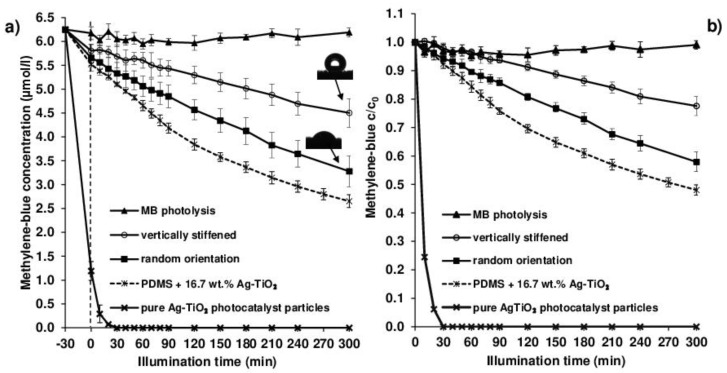
Absolute (**a**) and relative (**b**) representation of the LED light induced photodegradation of 2 mg/L methylene blue (MB) model dye solution on the surface of 2.6 × 7.6 cm^2^ magnetoresponsive thin film (63.3 ± 1.6 mg/cm^2^ specific mass, 16.7 wt.% Ag-TiO_2_ content) with different grass orientations. The direct photolysis of MB solution, and the photocatalytic efficiencies of the pure Ag-TiO_2_ particles and the non-magnetic composite samples are also presented as references.

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
