# Peer review of "A Stimulus-Responsive Polymer Composite Surface with Magnetic Field-Governed Wetting and Photocatalytic Properties"

_polymers, 2020, doi:10.3390/polym12091890_

Round 1

Reviewer 1 Report

This is an interesting work, where the authors combine in a smart way magnetic responsiveness and photocatalytic efficiency.

The manuscript merits publication.

Some minor (or, less minor) remarks:

  1. The authors should be a little more careful when writing the manuscript. For example, some remarks concerning Fig. 1 are listed below:

Fig.1a) A little more discussion is needed, in order to follow the reaction scheme.

Fig. 1b) Please use different symbols for each mixture studied and define the percentage values shown.

Fig.1c) i) Unless there is a mistake, these results do not seem consistent with the results shown in Fig. 1b), ii) The deformation used to extract the values of G’/G’’ for this figure should be reported. iii) How the data for crosslinker content 0 and 100% were determined? Fig. 1c does not contain these results. iv) Is there a reason to use a linear y-axis in Fig.1c, whereas the respective axis in Fig. 1b is logarithmic?

Another example is section 4. Discussion.

       2. Some “blank” experiments would be useful. For example, what is the photocatalytic efficiency of composite material prepared in the absence of magnetic field or not-containing magnetic microparticles?

        3. While the combination of both properties is nicely explained and demonstrated, I wonder what would be the benefit of applying a magnetic-field in a real application, since photocatalytic efficiency decreases.

Author Response

We answered the Reviewer's questions. Please see the attachment.

Reviewer 2 Report

I have read the manuscript “Stimulus-responsive polymer composite surface with magnetic field-governed wetting and photocatalytic properties” by László Mérai et al. (MS # polymers-907184) submitted for the publication in Polymers.

The authors report the results on the preparation and characterization of polymer composites (carbonyl Fe and Ag-TiO2 loaded in PDMS) showing wettability and photocatalytic properties dependent on the applied magnetic field.

The paper is interesting and generally well written (even if there are some misprints to be corrected), but it is not clear to the referee the presence of section 4:

“ 4. Discussion

Authors should discuss the results and how they can be interpreted in perspective of previous studies and of the working hypotheses. The findings and their implications should be discussed the broadest context possible. Future research directions may also be highlighted.”

Maybe it is a typo, but it can be applied as a referee’s suggestion to the authors in order to further improve their manuscript. In addition:

  1. Figure 8.a and Figure b are equivalent; one of them can be withdrawn;
  2. Figure 2 shows an inset (magnetic properties of carbonyl Fe),which is not described in the caption;
  3. Figure 1: the use of powers of 10 is preferable as values of x and y axis.

Author Response

(The authors gave the same response as above.)
